# Acute MPTP Treatment Impairs Dendritic Spine Density in the Mouse Hippocampus

**DOI:** 10.3390/brainsci11070833

**Published:** 2021-06-23

**Authors:** Poornima D. E. Weerasinghe-Mudiyanselage, Mary Jasmin Ang, Mai Wada, Sung-Ho Kim, Taekyun Shin, Miyoung Yang, Changjong Moon

**Affiliations:** 1Department of Veterinary Anatomy, College of Veterinary Medicine and BK21 FOUR Program, Chonnam National University, Gwangju 61186, Korea; 208314@jnu.ac.kr (P.D.E.W.-M.); mcang3@up.edu.ph (M.J.A.); wataametokatatumuri@gmail.com (M.W.); shokim@chonnam.ac.kr (S.-H.K.); 2Department of Basic Veterinary Sciences, College of Veterinary Medicine, University of the Philippines Los Baños, Los Baños 4031, Philippines; 3Department of Veterinary Anatomy, College of Veterinary Medicine, Jeju National University, Jeju City 63243, Korea; shint@jejunu.ac.kr; 4Department of Anatomy, Wonkwang University School of Medicine, Iksan 54538, Korea

**Keywords:** acute MPTP-lesioned mouse model, dendritic complexity, dopaminergic system, Golgi staining, Parkinson’s disease, structural plasticity

## Abstract

Among the animal models of Parkinson’s disease (PD), the 1-methyl-4-phenyl-1,2,3,6-tetrahydropyridine (MPTP)-lesioned mouse model has shown both dopaminergic (DA) damage and related motor control defects, as observed in patients with PD. Recent studies have suggested that the DA system interacts with the synaptic plasticity of the hippocampus in PD. However, little is known about how alterations in the hippocampal structural plasticity are affected by the DA damage in MPTP-lesioned models. In the present study, we investigated alterations in dendritic complexity and spine density in the mouse hippocampus following acute MPTP treatment (22 mg/kg, intraperitoneally, four times/day, 2-h intervals). We confirmed that acute MPTP treatment significantly decreased initial motor function and persistently reduced the number of tyrosine hydroxylase-positive DA neurons in the substantia nigra. Golgi staining showed that acute MPTP treatment significantly reduced the spine density of neuronal dendrites in the cornu ammonis 1 (CA1) apical/basal and dentate gyrus (DG) subregions of the mouse hippocampus at 8 and 16 days after treatment, although it did not affect dendritic complexity (e.g., number of crossing dendrites, total dendritic length, and branch points per neuron) in both CA1 and DG subregions at all time points after treatment. Therefore, the present study provides anatomical evidence that acute MPTP treatment affects synaptic structure in the hippocampus during the late phase after acute MPTP treatment in mice, independent of any changes in the dendritic arborization of hippocampal neurons. These findings offer data for the ability of the acute MPTP-lesioned mouse model to replicate the non-nigrostriatal lesions of clinical PD.

## 1. Introduction

Parkinson’s disease (PD) is a common neurodegenerative disorder with a pathological hallmark of dopaminergic (DA) cell loss in the substantia nigra [1]. Several animal models have been used to explore the pathogenic pathways of PD, as well as the therapeutic mechanisms against it, because no single model is likely to be appropriate in all studies [2]. Amongst these, the 1-methyl-4-phenyl-1,2,3,6-tetrahydropyridine (MPTP)-lesioned rodent model is a classical systemic model that is used in the preclinical testing of therapies to improve symptoms, as well as in the screening of pharmacological and genetic therapies [2]. As major vulnerable brain regions in the DA system, the substantia nigra and striatum have been a focus of research into pathological changes in PD. In both patients with PD and MPTP-lesioned models, striatal DA denervation appears to be related to the reduced dendritic length and spine number in this brain region [3,4,5,6]. In recent studies, MPTP administration altered synaptic plasticity in the hippocampus, suggesting that the DA system interacts with neuronal plasticity in the hippocampus [7,8]. 

Neurodegenerative diseases are pathological conditions characterized by progressive deterioration of synapses and neurons in specific anatomical locations [9]. For example, reduced dendritic spine count has been reported in the hippocampal neurons in animal models of Alzheimer’s disease [10], and dendritic abnormalities and spine pathology have been observed in both human patients and mouse models of Huntington’s disease, a fatal neurodegenerative condition characterized by cognitive, psychiatric, and motor symptoms [11,12,13]. Furthermore, non-medicated and non-demented patients with early PD have atrophy in the prefrontal cortex and hippocampus [14].

However, the neuronal architecture (i.e., dendritic and spine structure) of the hippocampus in PD animal models, specifically the MPTP-lesioned mouse model, has not been yet clarified. In the present study, we concentrated on the temporal profile of neuronal micromorphometry in mouse hippocampal *cornu ammonis* 1 (CA1) and dentate gyrus (DG) neurons following acute MPTP treatment. To this end, we used Golgi impregnation, a classical approach for staining and counting dendritic spines [15]. Our data offers new insight into the extra-nigrostriatal pathologies, specifically regarding alterations in the structural plasticity of hippocampal neurons in the acute MPTP-lesioned mouse model.

## 2. Materials and Methods

### 2.1. Animals 

Twelve-week-old male C57BL/6N mice weighing 25–28 g were obtained from Daihan-Biolink Co. (Chungbuk, Korea). They were acclimatized and quarantined for 1 week prior to use in experiments. The animals were maintained in an area sustained at a temperature of 23 °C ± 2 °C, a relative humidity of 50% ± 5%, a 12 h light–dark cycle, and 13–18 changes in air volume per hour. Mice were given ad libitum access to commercial rodent chow (Jeil Feed Co., Daejeon, Korea) and water. The procedures and protocols adopted in the present study were in accordance with the Institutional Animal Care and Use Committee of Chonnam National University (CNU IACUC-YB-2019-22), and animal care conformed to internationally agreed standards for laboratory animal use and care, as dictated by the National Institutes of Health (NIH) (NIH Publication No. 8023, revised 1978). Every effort was made to minimize the number of animals used and the discomfort of the animals in the experiments.

### 2.2. Experimental Design and MPTP (1-Methyl-4-Phenyl-1,2,3,6-Tetrahydropyridine) Treatment

In total, 52 mice were used in the current study, divided into two separate experiments, as shown in Figure 1A. Following acclimatization, mice were randomly split into two groups. The acute MPTP intoxication procedure was performed as described previously [16]. Briefly, the MPTP group received probenecid (250 mg/kg; Sigma-Aldrich, St. Louis, MO, USA) 30 min before the 1st MPTP injection to block accelerated clearance and excretion of MPTP from the brain and kidney after injection [17]. Next, four intraperitoneal injections of 22 mg/kg MPTP (Sigma-Aldrich) were administered at 2 h intervals within a single day, for an overall dose of 88 mg/kg. The dosage of MPTP was empirically derived to achieve a ~50% decrease in motor function. The control group received saline injections only. The motor function was assessed consecutively at 1, 2, 4, and 8 days (Experiment 1: *n* = 8 mice/group, total = 16 mice), and the animals were sacrificed at 1, 8, and 16 days after the last injection of MPTP or saline (Experiment 2: *n* = 6 mice/group at each time point, total = 36 mice). Brain samples were collected for immunohistochemistry (*n* = 6 hemispheres/group) and for Golgi impregnation (*n* = 6 hemispheres/group). The brain hemispheres collected for immunohistochemical analysis were fixed in 4% paraformaldehyde (*w*/*v*) in phosphate-buffered saline (PBS), while other hemispheres bound for Golgi staining were rinsed with 0.1 M phosphate buffer and immersed in Golgi–Cox solution (FD Neurotechnologies, Ellicott City, MD, USA). 

### 2.3. Rotarod Test

The rotarod test was performed following a previously reported protocol, with slight modifications [18,19]. The rotarod (Mouse Rota-Rod, Ugo Basile, Varese, Italy) was programmed to rotate with linearly increasing speed from 5 to 40 rpm in 300 s. When the animals fell off the rod, the time (s) and speed attained (rpm) were automatically calculated. The animals (*n* = 8 mice/group) were pre-trained for 2 days before MPTP intoxication. During pre-training, the mice performed three trials a day with 20 min inter-trial intervals. During each trial, the mice were conditioned at a speed of 5 rpm for 300 s. On the test day, an average of the speed attained and latency to fall were determined from three consecutive trials performed with 20 min intervals in one day.

### 2.4. Immunohistochemistry

All immunohistochemical procedures were carried out in accordance with previous reports [20]. The fixed brain hemispheres were merged in 30% (*w*/*v*) sucrose for 3–7 days. Thereafter, they were sectioned at 30 µm thickness in the coronal plane using a sliding microtome (SM2010R; Leica Microsystems, Wetzlar, Germany) and stored in PBS at 4 °C. Free-floating sections were incubated in 0.3% (*v*/*v*) hydrogen peroxide for 20 min to inactivate endogenous peroxidase activity, blocked with 5% (*v*/*v*) normal goat serum (Vector ABC Elite Kit; Vector Laboratories, Burlingame, CA, USA) in 0.3% (*v*/*v*) Triton X-100 for 1 h at room temperature (RT; 22 °C ± 2 °C). Sections were next incubated in rabbit anti-tyrosine hydroxylase (TH; 1:500; Millipore, Darmstadt, Germany) overnight at 4 °C. Following washing, sections were incubated for 1 h at RT with biotinylated goat anti-rabbit IgG (Vector ABC Elite Kit; Vector Laboratories), washed, and incubated for 1 h at RT with an avidin-biotin peroxidase complex (Vector ABC Elite Kit; Vector Laboratories) prepared as per the manufacturer’s specifications. After three washes with PBS, the peroxidase reaction was induced using a diaminobenzidine substrate (DAB kit; Vector Laboratories) prepared according to the manufacturer’s instructions. As a negative control, the primary antibody was omitted from a few test sections (data not shown). 

Images of immunohistochemical staining were acquired using a Leica microscope with Leica Caption Suite software (v4.12.0; Leica Microsystems CMS GmbH). The total number of TH-immunopositive cells in the substantia nigra pars were counted manually at 20× magnification using a Leica microscope (v4.12.0; Leica Microsystems CMS GmbH), as previously described [21]. TH-immunopositive cells, which were clearly demarcated from background staining, were counted by a blinded observer. Three hemisections of the substantia nigra, which lies approximately 3.64 mm caudal of the bregma, were selected from each animal for cell counting. The numbers of TH-immunopositive cells from the three non-overlapping sections (approximately 60 µm apart) were averaged per animal. The mean number of immunopositive cells in the three sections of each mouse was taken as *n* = 1. The number of TH-immunopositive cells per group was averaged and expressed as mean ± standard error (SE; *n* = 6 mice/group).

### 2.5. Golgi Staining

Hippocampal neuron arbor complexity and the number of dendritic spines were traced and quantified using Golgi impregnation, which was performed similarly to a method described previously [20,22]. The FD Rapid Golgistain^TM^ Kit and associated methodology were used (FD Neurotechnologies, Ellicott City, MD, USA). Brain hemispheres (*n* = 6/group) were washed in 0.1 M phosphate buffer, submerged in Golgi–Cox solution for 14 days, and then moved to a sucrose-containing solution at RT for 3–7 days. Brain sections 200 μm in thickness were adhered to gelatin-coated slides with a drop of sucrose solution. The slides were stored in the dark to air dry at RT for 3 days; they were then processed for Golgi impregnation, as described by the manufacturer of the FD Rapid Golgistain^TM^ Kit.

### 2.6. Hippocampal Neuron Tracing and Sholl Analysis

Both CA1 and DG subregion neurons from the hippocampus were traced and analyzed, as mentioned in previous reports [20]. Neurons that fulfilled the following criteria were chosen for the evaluation of dendritic arbor complexity: good-impregnation, unbroken branches, cell body located within the area of interest, and branches that were sufficiently separated from the neighboring neurons to minimize interference in the study. Ten randomly distributed neurons from each hippocampal region over three brain sections (approximately 2.12 mm behind bregma) in each animal were visualized at a magnification of 200× using a camera lucida. Dendrites were quantitatively evaluated in each of the selected neurons using Sholl’s concentric circle method [23]. A set of concentric circles that were spaced 10 µm apart was placed over the neurons and centered on the cell body; dendrites intersecting each circle were then counted to compute the number of dendritic intersections at different radial distances from the neuronal soma, total dendritic length, and branch points. Each value was averaged per mouse, and the mean value of each mouse was taken as *n* = 1. The value per group was averaged and expressed as mean ± SE (*n* = 6 mice/group).

### 2.7. Spine Density Measurement

Dendritic spine density was evaluated in the CA1 and DG subregions of the hippocampus using a Leica microscope with Leica Caption Suite software (v4.12.0; Leica Microsystems CMS GmbH). Dendritic spine distribution was analyzed based on constantly stained, unbranched, contiguous dendritic segments only. All conspicuous protrusive dendritic spines were counted from the terminal to the tip over 30 μm dendritic segments at a magnification of ×1500. Five neurons from each hippocampal region were selected randomly over three brain sections from each animal, and two dendritic segments from each neuron were selected at the level of secondary and tertiary branching. Thus, in total, 10 segments in each animal were counted, and spine density was derived as the number of spines per 10 µm of dendritic length [20]. The number of spines was averaged per mouse, and the mean number of each mouse was taken as *n* = 1. The number per group was averaged and expressed as mean ± standard error (SE, *n* = 6 mice/group).

### 2.8. Statistical Analysis

All statistical analyses were performed using Prism (GraphPad Software, San Diego, CA, USA). To test the main effects of treatment and time (or distance), as well as their interactions with results of TH immunoreactivities, behavioral test, Sholl analyses, and spine counting, two-way analysis of variance (ANOVA) followed by Šidák’s multiple comparison test was used. The results of the ANOVAs are presented in Appendix A. In all statistical tests, *p*-values < 0.05 were considered statistically significant. The sample sizes used per experiment are mentioned in the results section and figure legends. 

To calculate the minimum anticipated sample size per group, a power analysis was performed using La Morte’s power calculator at an alpha level of 0.05 to reach a power of at least 95% (https://www.bu.edu/researchsupport/compliance/animal-care/working-with-animals/research/sample-size-calculations-iacuc/ accessed on 27 May 2021). The following sample sizes were produced using this method as the minimum requirement: *n* = 3 mice for the rotarod test; *n* = 4 mice for the immunohistochemistry experiment’s relative count of TH-positive cells; and *n* = 6 mice for the Sholl analysis and dendritic length, branch point, and spine density. The absolute sample sizes applied in each experiment are included in the results section, figures, and legends. All sample sizes for each experiment were set to be greater than the minimum requirement from the results of power analysis.

## 3. Results

### 3.1. Acute MPTP Treatment Caused Impairment of Motor Function and Dopaminergic (DA) Cell Loss in Mice 

To validate the current MPTP-lesioned model, locomotor activity was assessed using the rotarod test (*n* = 8 mice/group). One day after injection, the rotation speed attained was significantly lower in the MPTP group than in the control group (control group: 34.08 ± 1.457 rpm; MPTP group: 16.58 ± 1.622 rpm; *p* < 0.0001; Figure 1B, left panel), as was the latency to fall (control group: 244.7 ± 13.44 s; MPTP group: 96.08 ± 13.16 s; *p* = 0.0002; Figure 1B, right panel). Similarly, at 2 days post-injection, MPTP group still attained a significant lower rotation speed, compared to those of the control group (control group: 35.12 ± 1.844 rpm; MPTP group: 25 ± 3.259 rpm; *p* = 0.0394; Figure 1B, left panel) and latency to fall (control group: 255.3 ± 15.73 s; MPTP group: 167.3 ± 27.63 s; *p* = 0.0447; Figure 1B, right panel). However, there were no observable differences thereafter in terms of rotation speed attained and latency to fall between the control and MPTP group. 

To confirm that MPTP lesioning had occurred in the brains, immunohistochemistry for TH expression in the substantia nigra was performed at different time points (1, 8, and 16 days). As shown in Figure 1C, the number of TH-immunopositive cells in the substantia nigra of the MPTP group was significantly lower than those of the control group at 1 (control group: 45.67 ± 3.127; MPTP group: 24.94 ± 3.834; *p* = 0.005), 8 (control group: 47.44 ± 2.841; MPTP group: 27.61 ± 4.317; *p* = 0.0074), and 16 days (control group: 54.28 ± 6.244; MPTP group: 35.56 ± 4.242; *p* = 0.0119) after treatment. This result indicates that the acute MPTP protocol used in the present study only induces motor impairment in the early stages after treatment, but persistently decreased DA cell count at all time points after treatment. 

### 3.2. Acute MPTP Treatment Did Not Alter Dendritic Complexity in the CA1 and DG Subregions of the Mouse Hippocampus

At 1, 8, and 16 days after the injection, the dendritic complexity of the CA1 and DG neurons was traced and quantified using Golgi staining. In the hippocampal CA1 subregion, the number of dendritic intersections at different radial distances from the neuronal soma was counted using Sholl analysis (Figure 2). There was no significant difference in the number of dendritic intersections in both CA1 apical and basal neurons between the control and MPTP groups at any radial distance from the soma (Figure 2C). Moreover, the total dendritic length (Figure 3A) and branch points per neuron (Figure 3A) in the CA1 subregion showed no significant difference between the control and MPTP groups at any time point examined. In the hippocampal DG subregion, dendritic complexity (the number of dendritic intersections, total dendritic length, and the branch point count per neuron) showed no significant difference between the control and MPTP-treated groups at any time point observed (Figure 4). 

### 3.3. Acute MPTP Treatment Significantly Reduced Dendritic Spine Density in the Hippocampal CA1 and DG Subregions

Acute MPTP treatment significantly reduced spine density in the hippocampal CA1 and DG neurons 8 and 16 days after injection (Figure 5). The total number of spines per 10 µm of neuronal dendrite in the CA1 apical region was significantly lower in the MPTP-treated group than in the control group at 8 days (control group: 15.36 ± 1.450, MPTP group: 10.59 ± 1.449, *p* = 0.0215) and 16 days (control group: 13.77 ± 1.397, MPTP group: 9.235 ± 0.365, *p* = 0.0298; Figure 5A) post treatment. Similarly, in the basal CA1, the dendritic spine density in the MPTP-treated group was significantly lower than that in the control group at 8 days (control group: 14.83 ± 0.619, MPTP group: 10.55 ± 1.700, *p* = 0.0364) and 16 days (control group: 13.84 ± 1.513, MPTP group: 9.004 ± 0.277, *p* = 0.0154; Figure 5B) post treatment. Moreover, in the DG, the spine density was significantly reduced in the MPTP-treated group at 8 days (control group: 16.73 ± 0.694, MPTP group: 11.55 ± 1.835, *p* = 0.015) and 16 days (control group: 14.64 ± 1.489, MPTP group: 10.01 ± 0.585, *p* = 0.0329; Figure 5C) post-treatment. 

## 4. Discussion

In the present study, we assessed temporal alterations in the neuronal architecture of the mouse hippocampus following acute MPTP treatment. Assessments were made on time points coinciding with the motor-impairment period on days 1 and 2 post-treatment (early phase) until the onset of motor recovery on days 8 and 16 post-treatment (late phase). However, prolonged and constant DA cell loss was observed despite recovery from motor impairments. Interestingly, acute MPTP treatment severely impaired spine densities in the hippocampal CA1 and DG subregions during the late phase after treatment, independent of any alterations in the dendritic complexity (e.g., number of crossing dendrites, total dendritic length, and branch points per neuron).

The pathological features of PD include DA neuron degeneration in the substantia nigra, leading to motor fluctuations [1]. Similar to previous studies [24,25], the present study confirmed an initial decrease in motor function, as well as prolonged decreases in the number of DA neurons in the substantia nigra, suggesting that the regimen used appropriately replicated symptoms of PD. In a number of neurodegenerative diseases (i.e., Alzheimer’s disease, schizophrenia, Down syndrome, and autism spectrum disorders) as well as in normal aging, the pathological hallmarks are related to alterations in the dendritic structure and/or spine density [26,27,28]. Previous studies have found that altered levels of neurochemicals, mainly dopamine, in the striatum trigger changes in the dendritic length and spine density of striatal neurons. This is primarily observed in the caudate putamen of patients with PD and MPTP-lesioned monkeys [3,5]. In the hippocampus, MPTP treatment also alters neurochemical levels and reduces neurogenesis [29,30]. The hippocampus and nigrostriatal system are involved in the non-motor features of PD, such as cognitive and emotional dysfunction [30,31,32]. Alterations in dendritic structure/spine density in the hippocampus are associated with neurological and neuropsychiatric diseases including mental retardation and dementia [33], possibly accounting for the non-motor impairments of PD. In the present study, we confirmed that acute MPTP treatment led to impaired dendritic spine density in the CA1 and DG subregions of the hippocampus in the late phase without any alterations in the dendritic complexity, suggesting that the changes of the dendritic spines in the hippocampus might be related to the non-motor, PD-like symptoms.

There are several possible mechanisms related to DA underlying this decrease in spine density after acute MPTP treatment in the hippocampus. Firstly, Janakiraman et al. [30] reported reduced dopamine and serotonin levels in the hippocampus after a chronic MPTP regimen. Moreover, dopamine facilitates dendritic proliferation, and DA depletion results in shorter dendrites and reduced spine density in the nucleus accumbens [34], supporting the possibility that prolonged DA depletion in the present study may change the spine density in the hippocampus at the late phase. Moreover, although DA is synthesized mainly by mesencephalic neurons, DA neurons project to the limbic system, including to the hippocampus. Thus, DA receptors are known to control functions related to cognition and synaptic plasticity in the hippocampus [35]. DA innervation in the hippocampus originated in the projections coming from the ventral tegmental area (VTA) [36,37], via this circuitry, DA neurons from the VTA can modulate the plasticity of hippocampal neurons [38]. Thus, acute MPTP may have targeted several possible circuits connecting the hippocampus and VTA, which may include the inhibitory connections of the lateral habenula and rostromedial tegmental nucleus to VTA neurons [39,40,41,42], and the excitatory connections of the pedunculopontine tegmental nucleus (PPTg) [43] and laterodorsal tegmentum (LDT) [44] to VTA neurons. This may consequently alter the synaptic plasticity in the hippocampus through changes in the dendritic spine morphology. Further confirmatory studies exploring the roles of these hippocampal-VTA circuitries are needed to uncover the possible underlying mechanisms that lead to the spine density impairment induced by acute MPTP treatment. Moreover, these circuitries may also offer mechanisms by which acute MPTP can elicit non-motor behavioral impairments reminiscent of clinical PD. Furthermore, deficiency in dopamine transporter (DAT) have also resulted to decreased spine densities in the hippocampal CA1 subregion mice [45]. This provides another DA-related mechanism by which acute MPTP altered synaptic densities in the CA1 and DG subregions.

Although present, the DA innervation of the hippocampus is moderate; thus, other non-DA causes of the impairment in synaptic densities must be addressed. For instance, cholinergic signaling may have a role in the synaptic morphological alterations observed, as MPTP treatment have also been reported to induce downregulation of cholinergic markers in the septohippocampal system [46]. These cholinergic deficits, in turn, have been found to reduce dendritic branching and spine density in the neocortical pyramidal neurons of rats [47]. Thus, further studies are needed to verify if cholinergic signaling is involved in the synaptic impairments observed in the current study. 

## 5. Conclusions

In the present study, the acute MPTP regimen caused DA cell loss and marked motor deficits in lesioned mice, consistent with many previous studies. Furthermore, acute MPTP treatment altered dendritic spine density in the hippocampus during the late phase post-treatment (motor recovery). Taken together, these changes may have been interrelated or controlled separately through multiple mechanisms, and future studies should address this distinction. These new findings suggest that inhibition of the DA system results in changes to the neuronal architecture of the hippocampus during the late phase after MPTP treatment, providing anatomical evidence that structural plasticity in hippocampal neurons may be involved in the etiology of the non-motor hallmarks of PD. Moreover, the validity of the acute MPTP model in replicating the structural alterations present in the hippocampus of clinical PD is confirmed by the results of the current study. 

## Figures and Tables

**Figure 1 brainsci-11-00833-f001:**
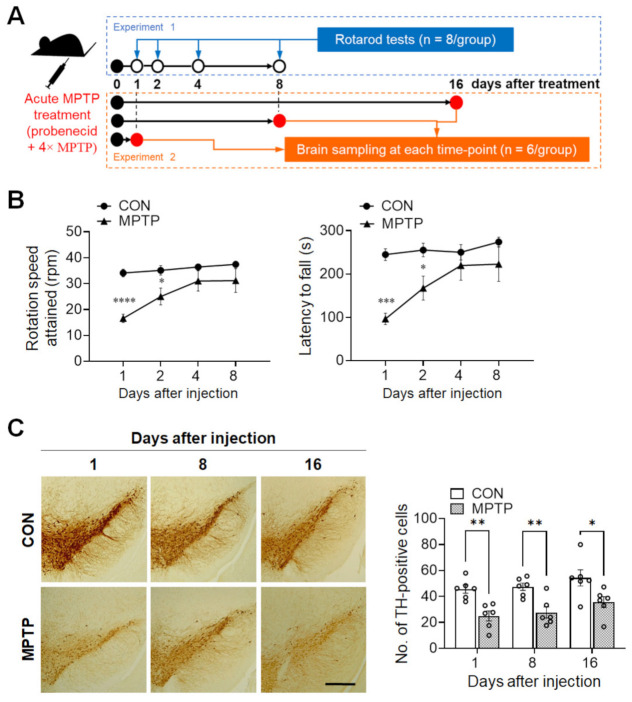
Acute MPTP (1-methyl-4-phenyl-1,2,3,6-tetrahydropyridine) treatment triggered initial motor impairment and persistently decreased tyrosine hydroxylase (TH) immunoreactivity in the substantia nigra. (**A**) Schematic representation of the experimental design. In experiment 1, the rotarod test was performed at 1, 2, 4, and 8 days after acute MPTP injection (4 × 22 mg/kg, 2 h intervals). In experiment 2, mice were sacrificed at 1, 8, and 16 days post-injection, and brain samples were collected for immunohistochemical analysis and Golgi staining. (**B**) Motor function was evaluated using the rotarod test. Early phase (at 1 and 2 days) after MPTP treatment, mice showed a significant decrease in the rotation speed attained and latency to fall in the early phase. (**C**) TH immunoreactivity in the substantia nigra was significantly lower at 1, 8, and 16 days after injection in the MPTP group than in the control (CON) group. Data are expressed as mean ± standard error (*n* = 8 mice/group for rotarod; *n* = 6 hemispheres/group at each time point for the TH immunoreaction). The scale bar (**C**) represents 200 µm. * *p* <0.05, ** *p* < 0.01, *** *p* < 0.001, **** *p* < 0.0001 between the CON and MPTP-treated groups.

**Figure 2 brainsci-11-00833-f002:**
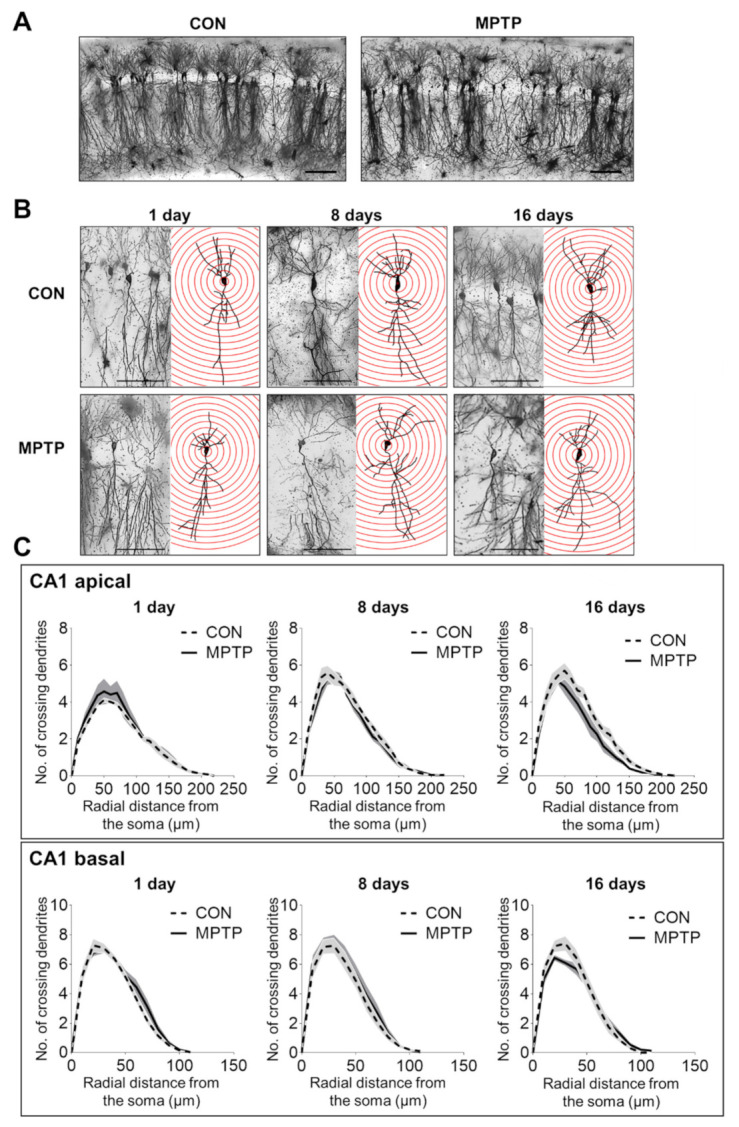
Golgi staining revealed the dendritic arborization of the hippocampal *cornu ammonis* 1 (CA1) subregion following acute MPTP treatment. (**A**) Representative images of CA1 subregion of control (CON) and MPTP-treated group at lower magnification. (**B**) Representative images of CA1 pyramidal neurons from the CON and MPTP-treated groups at 1, 8, and 16 days after injection at higher magnification. (**C**) Line graphs show the number of intersections per 10 µm radial unit distance from the soma in apical (upper panel) and basal (lower panel) dendrites at 1, 8, and 16 days after injection. Data are expressed as the mean ± standard error (*n* = 6 mice/group). The scale bars in A and B represent 200 µm and 50 µm, respectively.

**Figure 3 brainsci-11-00833-f003:**
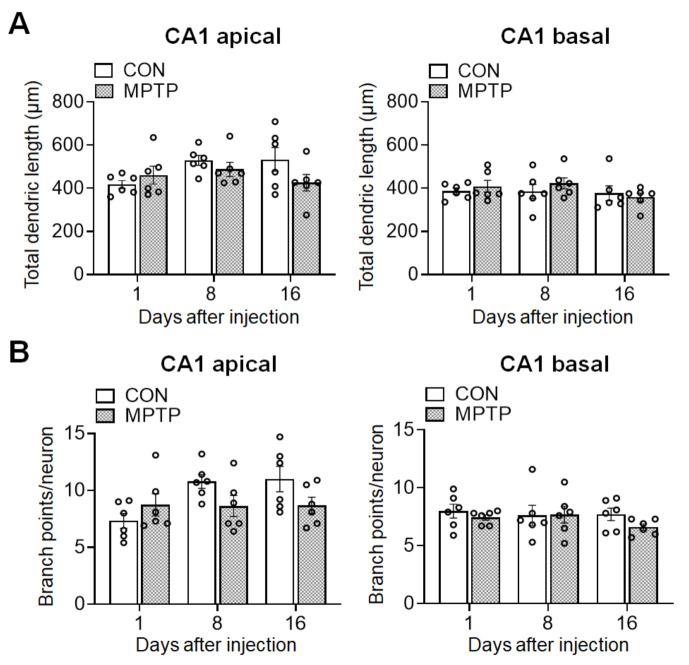
Acute MPTP treatment did not alter the dendritic length and branch points in *cornu ammonis* 1 (CA1) pyramidal cells. The bar graphs represent the total dendritic length (**A**) and branch point counts (**B**) in the apical (left panels) and basal (right panels) CA1 of the control (CON) and MPTP-treated groups. Data are expressed as the mean ± standard error (*n* = 6 mice/group).

**Figure 4 brainsci-11-00833-f004:**
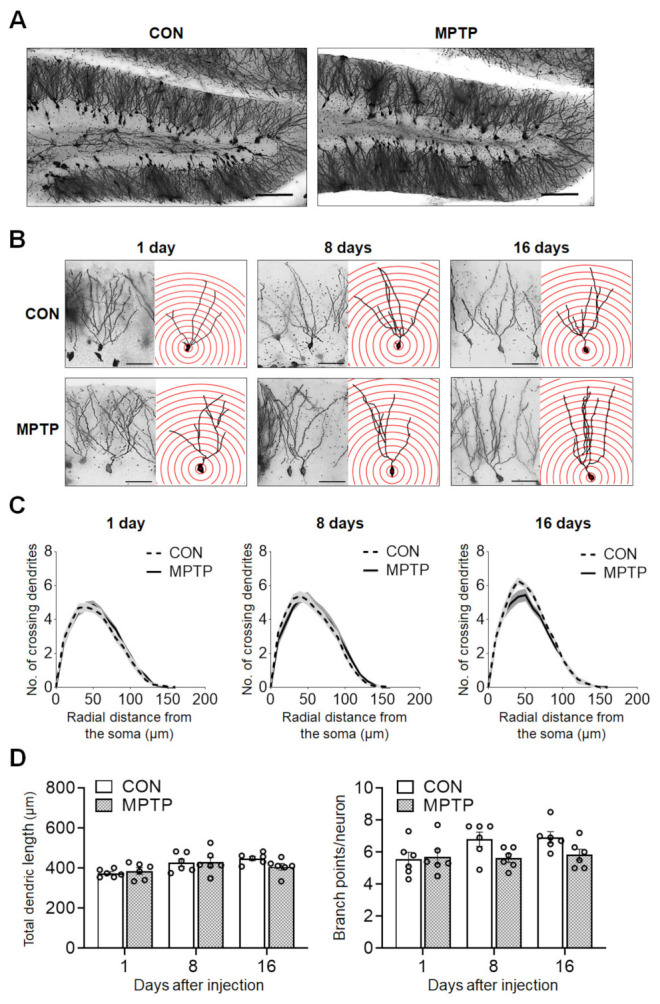
Acute MPTP treatment did not change dendritic complexity in dentate gyrus (DG) granular neurons. Golgi staining of brains from the control (CON) and MPTP groups revealed comparable dendritic complexity in the DG region. (**A**) Representative images of the DG subregion of CON and MPTP-treated group at lower magnification. (**B**) Representative images of DG granular neurons from the CON and MPTP-treated groups at 1, 8, and 16 days after injection at higher magnification. (**C**) Line graphs show the mean number of dendrite intersections per 10 µm radial unit distance from the soma. (**D**) Bar graphs display the total dendritic length (left panel) and dendritic branch point count (right panel). Data are expressed as the mean ± standard error (*n* = 6 mice/group). The scale bars in A and B represent 100 µm and 25 µm, respectively.

**Figure 5 brainsci-11-00833-f005:**
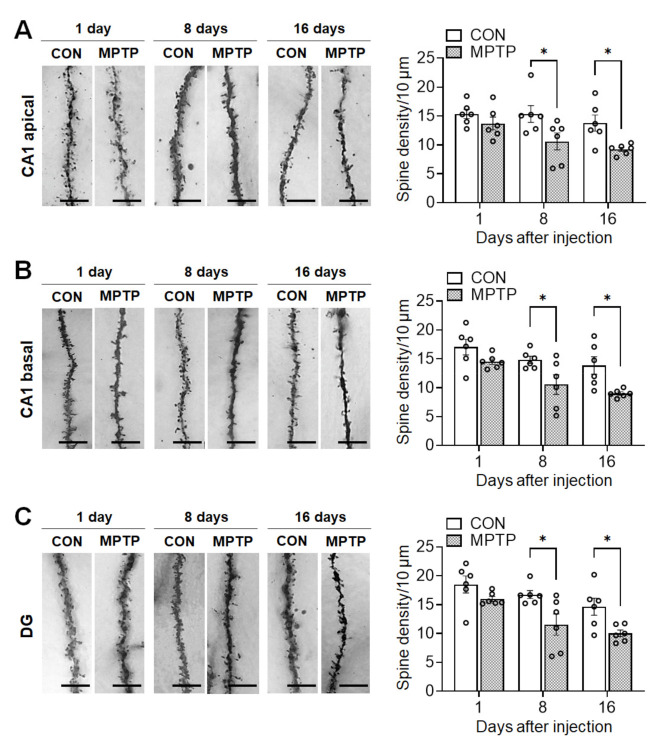
Acute MPTP treatment altered dendritic spine density in the hippocampus. Representative images from each group display dendritic spines along Golgi-impregnated neurons in *cornu ammonis* 1 (CA1) apical dendrites (**A**, left panel), a CA1 basal dendrites (**B**, left panel) and dentate gyrus (DG) dendrites (**C**, left panel). Bar graphs (**A**–**C**, right panel) display the spine density per 10 µm of dendrite. Data are expressed as the mean ± standard error (*n* = 6 mice/group). * *p* < 0.05 between the control (CON) and MPTP-treated groups. The scale bars in the dendritic images represent 5 µm.

## Data Availability

This paper utilized original data not used in other publications. The datasets generated and/or analyzed in the present study are available from the corresponding author upon reasonable request.

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
