# Peer review of "Acute MPTP Treatment Impairs Dendritic Spine Density in the Mouse Hippocampus"

_brainsci, 2021, doi:10.3390/brainsci11070833_

Round 1

Reviewer 1 Report

In their work, Weerasinghe-Mudiyanselage at all addressed whether acute MPTP treatment affects hippocampal neuronal architecure in the CA1 and DG. They measured dendritic complexity and spine density from Golgi-stained neurons of mice. In general, it is a clinically important finding, and can partially explain hippocampal atrophy in PD patients (Brück et al, 2003 - please cite).

Major comments:

- tThe experimental design is not exactly clear. wWhy the mice from experiment 1 were not usedd for the histological analysis, and only for behavioural tests?  What happened to the brain of those mice?

- Fig2: what is the reason of the short time effect on the number of crossings? How can it be altered in on the 1st day, but not on the 8th day?

-Fig3: The total dendritic length and branch points has a tendency to increase in the control group in the CA1 apivcal, but not in the basal part. Decrease in the MPTP group, in contrast, seems to be rather small between day1 and day16. Therefore, the differences between control and MPTP on day 16 seems to be rather the effect of these two (non-significant?), but opposing effects, than a "real" decrease in the MPTP group. Here the n=4/animal-based data would be especially helpful, instead of presenting and comparing 40 cells/group (again: the 10 cells/animal are not independent). Branch points in CA1 apical look more or less unchanged in the MPTP group, but increasing in the control group over time. What is the explanation for that?

-Statistical analysis needs substantial improvement (see below).

-1 day after MPTP treatment is rather short to see the effects of DA-depletion on morphology of the dendritic architecture: this normally takes normally longer time. Please discuss this, and provide plausible explanation, how can the observed changes be related to DA levels, or if not, to what?

Others:

Introduction:

-tThe reference list lacks recent citations - E.g. a paper from 2020 (Witzig et al., 2020) shows partially reversible changes in dendritic morphology in the striatum.

Methods:

-What was the rationale of using probenecid in addition to MPTP? Majority of the studies use only MPTP, and the DAergic lesion (Fig1B) is not convincingly higher than without. Probenecid itself is toxic (see later).

-Why only 3 mice were used for TH histology, and only 4 mice for Golgi? Did the authors make power analysis to determine the group size? Because of the technical limitations, 4 mice/group is a rather small group size for the dendritic morphological analysis. Analysis of the other 8 mice from Exp1 (fixed at day 8) would have provided better statistical outcome.

-Rotarod test: the overall distance was determined in three consecutive experiments or trials?

-Immunohistochemistry: using cobalt or nickel intensification might have provided better signal to noise ratio for the density analysis.

-TH analysis: which magnification was used to capture the images for the OD measurements? How many sections per animal were used for doing that? Based on the text, it was only one single section (3,64 from bregma). It is not reliable at all. The SN sections showed on Fig1B are not in the same bregma level – therefore, OD cannot be directly compared. Determining OD from the SN is not a common way of determining lesion anyway. Please, use either stereology, or at least manually count dopaminergic cell numbers from the entire SN (e.g. from every 5th section). It can be done using the existing, DAB-stained sections.

-Scholl analysis: please put the concentric circles over the neurons.

-Spine density measurement: it is not exactly clear, from where did the authors measure spine density? Between the first and second branchings? Or randomly selected segments? If latter, did they do a preliminary experiment to determine, how many cells have to be analyzed to compensate for the random effect? The spine density changes over the dendrites. Did those segments belong to different neurons? How the 10 segment/animal were chosen: how many brain sections and how many imaging field/brain sections were used? Similarly, the 10 neurons for Scholl analysis were selected from how many brain sections and imaging fields?

-Statistics: the authors analyzed 40 neurons or 40 dendritic segments/group from a total of 4 animals/group. As the 10 neurons or 10 dendritic segments from the same animal are not independent samples, they cannot be "treated" as n=10, rather averaging the results from those 10 neurons/segments, and then n=4 based on the 4 animals. The brain sections, imaging fields and neurons/segments can be incorporated in the analysis as random factors (mixed effect model).

Results:

-Fig1: it is interesting that the authors found significant effect of MPTP only on day 1, but not later. How can it be excluded, that this early effect is not due to other side-effects of the drugs (both MPTP and probenecid)? The mice are normally very weak and lose weight after MPTP injection, and probenecid lowers cellular ATP levels.: there is no big difference between control and MPTP, visible on the IHC images at days 1 and 16 - probably it is because not the same bregma levels are shown. Please replace the images.

-Fig2: quality of some Golgi images can be higher - e.g. control 16 days, MPTP 8 days. Please provide low magnification images of the hippocampus showing general impregnation quality in the groups. It would also help the readers if statistical significance was presented on the graph.

-Fig4.: please provide lower magnification, overview images, and higher quality images for the high magnification. In Panel C the dots are scattered, on panel D aligned.

-Fig5: Please replace some of the non-focused or lower quality images with better ones (e.g. CA1 basal, MPTP, day8). Spine density decreased in the MPTP group at day 8 in the CA1 apical, but on day 16 its mean seems to be "restored" to day 1, but the control increased. Were here the mice that were tested in the rotarod experiment used? Same for CA1 basal and DG.

Discussion

Dopaminergic innervation of hippocampus is moderate, therefore the rapid changes (within 1 day) is hard to be explained by DA depletion without proper examination - e.g. minimum DA levels by HPLC should be presented and correlated. On the long run, it might be a plausible explanation. Neurons in the HC have most probably very low DAT expression, as this protein is typical for the dopaminergic neurons. Therefore, changes directly due to MPTP is are also not very likely, but they needs further provident. On the other hand, cholinergic innervation of the HC might be decreased after MPTP treatment (Szegö et al., 2013), that can induce long-term changes in the dendritic architecture. Yet, this still does not explain day 1 differences. Changes in the spine density usually occurs earlier than the dendritic complexity, and both have distinct mechanisms. However, the common is that both take longer time.

In addition, although probenecid was found to be neuroprotective in some animal models (e.g. Vamos et al, 2009), probenecid lowers intracellular ATP levels, and can be neurotoxic (Alvarez-Fischer et al., 2013). Therefore, the short-time effect seen in this study might also be attributed to probenecid, and not to DA-dependent effects. Please, discuss this possibility.

Author Response

In their work, Weerasinghe-Mudiyanselage at all addressed whether acute MPTP treatment affects hippocampal neuronal architecture in the CA1 and DG. They measured dendritic complexity and spine density from Golgi-stained neurons of mice. In general, it is a clinically important finding, and can partially explain hippocampal atrophy in PD patients (Brück et al, 2004 - please cite).

Response: We sincerely appreciate the reviewer’s comment. We have cited the reference in the Introduction section, as follows:

(Lines 61–62)

“Furthermore, non-medicated and non-demented patients with early PD have atrophy in the prefrontal cortex and hippocampus [14].”

Major points

Comment 1. The experimental design is not exactly clear. Why the mice from experiment 1 were not used for the histological analysis, and only for behavioral tests? What happened to the brain of those mice?

Response: We sincerely appreciate the reviewer’s comment. As commented, the animals from experiment 1 were only used for the validation of PD-like motor dysfunctions following the present MPTP regimen. There is consensus that the exercise can induce neurogenesis and affect dendritic micromorphometry in the hippocampus (Redila & Christie, 2006; Stranahan, Khalil, & Gould, 2007). Thus, we did not use the animals from experiment 1 for histological and subsequent Sholl analysis to restrict any over-reaching behavioral effects. That is, we used the separated cohorts to evaluate the effects of the MPTP regimen on neuronal micromorphometry alone. We have revised the manuscript (Materials and Methods section, and Fig. 1A) to limit the readers’ confusion and provided more detailed information for separate cohorts, as follows:

(Lines 96–104)

“The motor function was assessed consecutively at 1, 2, 4, and 8 days (Experiment 1: n = 8 mice/group, total = 16 mice), and the animals were sacrificed at 1, 8, and 16 days after the last injection of MPTP or saline (Experiment 2: n = 6 mice/group at each time point, total = 36 mice). Brain samples were collected for immunohistochemistry (n = 6 hemispheres/group) and for Golgi impregnation (n = 6 hemispheres/group). The brain hemispheres collected for immunohistochemical analysis were fixed in 4% paraformaldehyde (w/v) in phosphate buffered saline (PBS), while other hemispheres bound for Golgi staining were rinsed with 0.1 M phosphate buffer and immersed in Golgi–Cox solution (FD Neurotechnologies, Ellicott City, MD, USA).”

Comment 2. What is the reason of the short time effect on the number of crossings? How can it be altered in on the 1st day, but not on the 8th day?

Response: We sincerely appreciate the comment. As commented regarding the Methods sections, we have increased the sample size as appropriate to the results from the power analysis computation and re-analyzed the dendritic complexity using more appropriate statistical methods. Unfortunately, the resulting data from re-analyses did not show any time point effects in the number of crossings. Thus, we have replaced Figures 2–4 and revised the results of dendritic complexity in the CA1 and DG subregions in the Results section of the revised manuscript, as follows:

(Lines 244–257)

3.2. Acute MPTP treatment did not alter dendritic complexity in the CA1 and DG subregions of the mouse hippocampus

At 1, 8, and 16 days after the injection, the dendritic complexity of the CA1 and DG neurons was traced and quantified using Golgi staining. In the hippocampal CA1 subregion, the number of dendritic intersections at different radial distances from the neuronal soma was counted using Sholl analysis (Fig. 2). There was no significant difference in the number of dendritic intersections in both CA1 apical and basal neurons between the control and MPTP groups at any radial distance from the soma (Fig. 2C). Moreover, the total dendritic length (Fig. 3A) and branch points per neuron (Fig. 3A) in the CA1 subregion showed no significant difference between the control and MPTP groups at any time point examined. In the hippocampal DG subregion, dendritic complexity (the number of dendritic intersections, total dendritic length, and the branch point count per neuron) showed no significant difference between the control and MPTP-treated groups at any time point observed (Fig. 4).”

Comment 3. The total dendritic length and branch points has a tendency to increase in the control group in the CA1 apical, but not in the basal part. Decrease in the MPTP group, in contrast, seems to be rather small between day1 and day16. Therefore, the differences between control and MPTP on day 16 seems to be rather the effect of these two (non-significant?), but opposing effects, than a "real" decrease in the MPTP group. Here the n=4/animal-based data would be especially helpful, instead of presenting and comparing 40 cells/group (again: the 10 cells/animal are not independent). Branch points in CA1 apical look more or less unchanged in the MPTP group, but increasing in the control group over time. What is the explanation for that?

Response: We sincerely appreciate the reviewer’s comment. As commented with regard to the statistical analyses, we have re-analyzed the results of all experiments regarding Sholl analysis and dendritic architectures using the increased sample size and comparative methods from each animal-based data. Unfortunately, there were no significant differences in any of the hippocampal subregions at any time point in terms of total dendritic length and the branch points per neuron. Thus, we have revised the Results section, as shown in the response to comment 2.

 Comment 4: Statistical analysis needs substantial improvement (see below).

Response: We sincerely appreciate the reviewer’s comment. As stated in the response to comments 2 and 3, we have recomputed all data in accordance with the methods suggested by the reviewer. We have extensively edited the manuscript to reflect these changes in the Materials and Methods, Results, and Figures sections. Furthermore, we have added more detailed information for statistical analyses, as follows:

(Lines 198–207)

“To calculate the minimum anticipated sample size per group, a power analysis was performed using La Morte's power calculator at an alpha level of 0.05 to reach a power of at least 95% (https://www.bu.edu/researchsupport/compliance/animal-care/working-with-animals/research/sample-size-calculations-iacuc/). The following sample sizes were produced using this method as the minimum requirement: n = 3 mice for the rotarod test; n = 4 mice for the immunohistochemistry experiment's relative count of TH-positive cells; and n = 6 mice for the Sholl analysis and dendritic length, branch point, and spine density. The absolute sample sizes applied in each experiment are included in the results section, figures, and legends. All sample sizes for each experiment were set to be greater than the minimum requirement from the results of power analysis.”

Comment 5: 1 day after MPTP treatment is rather short to see the effects of DA-depletion on morphology of the dendritic architecture: this normally takes normally longer time. Please discuss this, and provide plausible explanation, how can the observed changes be related to DA levels, or if not, to what?

Response: As stated above, we have made efforts to improve the statistical analyses and have increased the number of animals (n = 6) for morphometric experiments (thus total number of analyzed neurons = 60) in the revised manuscript. As observed in the re-analyzed data, there was no short-term effect of MPTP on dendritic architecture. Thus, we could not observe the DA-depletion effects on the dendritic architecture in the revised manuscript. We hope that the reviewer agrees with these implications. Also, we have modified the Figures, Abstract, Results, and Discussion sections, to reflect these facts.

 Others

Introduction:

  1. The reference list lacks recent citations - E.g. a paper from 2020 (Witzig et al., 2020) shows partially reversible changes in dendritic morphology in the striatum

Response: We agree with the comment and have included the suggested recent reference in the revised manuscript as follows:

(Lines 50–52)

“In both patients with PD and MPTP-lesioned models, striatal DA denervation appears to be related to the reduced dendritic length and spine number in this brain region [3-6].” (6th reference)

Methods:

  1. What was the rationale of using probenecid in addition to MPTP? Majority of the studies use only MPTP, and the DAergic lesion (Fig1B) is not convincingly higher than without. Probenecid itself is toxic (see later).

Response: We appreciate the reviewer’s comment. Probenecid has been used as an adjuvant with MPTP injection in mice to potentiate neurotoxicity by reducing the clearance of MPTP and its metabolites from the brain and kidneys (Petroske, Meredith, Callen, Totterdell, & Lau, 2001). MPTP/probenecid protocols have already been proven to potentiate MPTP neurotoxicity in mice, resulting in a more pronounced and prolonged DA depletion than with an MPTP-only protocol (Lau, Trobough, Crampton, & Wilson, 1990). Moreover, probenecid alone has no direct effect on striatal DA content (Lau et al., 1990). The injection regimen used in the present study was empirically derived through pilot studies to achieve a 50–80% survival rate while attaining ~50% motor dysfunction. Thus, we chose the combined regimen as the appropriate method for the present study.

  1. Why only 3 mice were used for TH histology, and only 4 mice for Golgi? Did the authors make power analysis to determine the group size? Because of the technical limitations, 4 mice/group is a rather small group size for the dendritic morphological analysis. Analysis of the other 8 mice from Exp1 (fixed at day 8) would have provided better statistical outcome.

Response: We sincerely appreciate the reviewer’s comment. Initially, we made all possible efforts to minimize the number of animals used while maintaining the validity and standards of the study. However, we also agree with reviewer’s comment. Thus, as mentioned above, we have increased sample size and re-analyzed our data according to the results of the power analyses. As a result, we have the increased sample size (n = 6) for the TH-staining and Sholl analysis and revised our manuscript to provide the mentioned information.

  1. Rotarod test: the overall distance was determined in three consecutive experiments or trials?

Response: We appreciate the reviewer’s comment. On the test day for rotarod, mice were tested on the rod programmed to rotate with linearly increasing speed from 5 to 40 rpm in 300 s. The average of the speed attained and latency to fall were determined from three consecutive trials with 20-min inter-trial intervals. To avoid confusion, we have made some changes as follows:

(Lines 113–115)

“On the test day, an average of the speed attained and latency to fall were determined from three consecutive trials performed with 20-min intervals in one day.”

  1. Immunohistochemistry: using cobalt or nickel intensification might have provided better signal to noise ratio for the density analysis.

Response: We appreciate the reviewer’s comment. However, ordinary immunohistochemical protocols that resulted in a brown color was successfully used for the detection of the tyrosine hydroxylase immunoreaction of the brain in previous studies (Ferraz et al., 2003; Xavier et al., 2005). Moreover, we observed an adequate number of immunopositive signals for TH using the general method. As the reviewer suggested below, we have changed the methods of comparison from relative density to immunopositive cell counting to improve the accuracy of the immunoreaction, with exclusion of noise signals. We have changed the text regarding TH staining in revised manuscript.

 TH analysis: which magnification was used to capture the images for the OD measurements? How many sections per animal were used for doing that? Based on the text, it was only one single section (3,64 from bregma). It is not reliable at all. The SN sections showed on Fig1B are not in the same bregma level – therefore, OD cannot be directly compared. Determining OD from the SN is not a common way of determining lesion anyway. Please, use either stereology, or at least manually count dopaminergic cell numbers from the entire SN (e.g. from every 5th section). It can be done using the existing, DAB-stained sections.

Response: We appreciate the reviewer’s comment. For analyzing TH immunoreaction, we used three sections approximately 3.64 mm caudal to the level of the bregma from each brain. For a more accurate comparison, we have analyzed the increased sample size (n = 6) for TH staining. Furthermore, we manually counted the TH immunopositive cells in the substantia nigra and replaced the representative images in Figure 1C to provide a better comparison, as suggested.

We have revised the text regarding TH staining to offer more detailed information as follows:

(Lines 135–145)

“The total number of TH-immunopositive cells in the substantia nigra pars were counted manually at 20× magnification using a Leica microscope (v4.12.0; Leica Microsystems CMS GmbH), as previously described [21]. TH-immunopositive cells, which were clearly demarcated from background staining, were counted by a blinded observer. Three hemisections of the substantia nigra, which lies approximately 3.64 mm caudal of the bregma, were selected from each animal for cell counting. The numbers of TH-immunopositive cells from the three non-overlapping sections (approximately 50 µm apart) were averaged per animal. The mean number of immunopositive cells in the three sections of each mouse was taken as n = 1. The number of TH-immunopositive cells per group was averaged and expressed as mean ± standard error (SE; n = 6 mice/group).”

  1. Scholl analysis: please put the concentric circles over the neurons.

Response: We have made the suggested change.

  1. Spine density measurement: it is not exactly clear, from where did the authors measure spine density? Between the first and second branchings? Or randomly selected segments? If latter, did they do a preliminary experiment to determine, how many cells have to be analyzed to compensate for the random effect? The spine density changes over the dendrites. Did those segments belong to different neurons? How the 10 segment/animal were chosen: how many brain sections and how many imaging field/brain sections were used? Similarly, the 10 neurons for Scholl analysis were selected from how many brain sections and imaging fields?

Response: We appreciate the reviewer’s comment. As mentioned above, we used six animals per group for the measurement of spine density. From each animal, five randomly distributed neurons from each hippocampal region were selected over three sections. Then, two dendritic segments from each neuron were selected at the level of secondary or tertiary branching. In total, ten dendritic segments were counted in each animal. For spine analysis, only intact, sufficiently impregnated, and unbranched segments were selected. All identifiable protruding dendritic spines were counted from the terminal to the tip at ×1500 magnification.

Similar to the selection of neurons for spine analysis, for Sholl analysis, ten randomly distributed neurons from each hippocampal region were also selected over three brain sections from each animal.

We have added more detailed information, as follows:

(Lines 182–188)

“Five neurons from each hippocampal region were selected randomly over three brain sections from each animal, and two dendritic segments from each neuron were selected at the level of secondary and tertiary branching. Thus, in total, ten segments in each animal were counted, and spine density was derived as the number of spines per 10 µm of dendritic length [20]. The number of spines was averaged per mouse, and the mean number of each mouse was taken as n = 1. The number per group was averaged and expressed as mean ± SE (n = 6 mice/group).”

(Lines 164–174)

“Ten randomly distributed neurons from each hippocampal region over three brain sections (approximately 2.12 mm behind bregma) in each animal were visualized at a magnification of 200× using a camera lucida. Dendrites were quantitatively evaluated in each of the selected neurons using Sholl's concentric circle method [23]. A set of concentric circles that were spaced 10 µm apart was placed over the neurons and centered on the cell body; dendrites intersecting each circle were then counted to compute the number of dendritic intersections at different radial distances from the neuronal soma, total dendritic length, and branch points. Each value was averaged per mouse, and the mean value of each mouse was taken as n = 1. The value per group was averaged and expressed as mean ± SE (n = 6 mice/group).”

  1. Statistics: the authors analyzed 40 neurons or 40 dendritic segments/group from a total of 4 animals/group. As the 10 neurons or 10 dendritic segments from the same animal are not independent samples, they cannot be "treated" as n=10, rather averaging the results from those 10 neurons/segments, and then n=4 based on the 4 animals. The brain sections, imaging fields and neurons/segments can be incorporated in the analysis as random factors (mixed effect model).

Response: We agree with the reviewer’s comment. We performed statistical analyses for Sholl data following many previous studies (Ang, Kang, & Moon, 2020; Flores et al., 2005; Kang et al., 2018; Sánchez, Gómez‐Villalobos, Juarez, Quevedo, & Flores, 2011). However, we agree that each neuron/dendritic segment was not from independent samples; thus, a mixed-effect model style of analysis might provide a better fit for our data. Therefore, we have re-analyzed the results using a mixed model effect while also adjusting the number of subjects per group to n = 6. We have edited the Materials and Methods and the Results and Discussion sections to reflect these changes.

Results:

  1. Fig1: it is interesting that the authors found significant effect of MPTP only on day 1, but not later. How can it be excluded, that this early effect is not due to other side-effects of the drugs (both MPTP and probenecid)? The mice are normally very weak and lose weight after MPTP injection, and probenecid lowers cellular ATP levels. There is no big difference between control and MPTP, visible on the IHC images at days 1 and 16 - probably it is because not the same bregma levels are shown. Please replace the images.

Response: We appreciate the reviewer’s comment. As commented, MPTP injection could induce several toxic effects reminiscent of clinical PD, including motor dysfunction, striatal dopaminergic neuron denervation, and other diverse neurological symptoms; thus, this drug has been widely used for inducing classical PD in animal models from previous studies. In Figure 1, we performed the rotarod test to confirm the mimic effects of MPTP to PD, resulting in the early effects of the drugs. To make the early effects clear, we have added an additional early time point (day 2) and revised the graph type to show clearer comparative results between the groups from the rotarod test. We hope the reviewer agrees that the significant differences during the early time point is due to the effects of drugs per se and not the side-effects.

Again, we have changed the sample size and statistical method for morphological analyses. Significant differences for TH immunoreactivity were still observed at all time points examined. We agree with the reviewer’s comment regarding the confusion with IHC images appearing to be on different levels; thus, we have changed the IHC image to show similar brain levels in Figure 1C of the revised manuscript.

  1. Fig2: quality of some Golgi images can be higher - e.g. control 16 days, MPTP 8 days. Please provide low magnification images of the hippocampus showing general impregnation quality in the groups. It would also help the readers if statistical significance was presented on the graph.

Response: As suggested, we have revised Figure 2. Unfortunately, there are no significant differences at any radial distance from the soma between the two groups even after analyzing with the newly updated sample size.

  1. Fig 4. Please provide lower magnification, overview images, and higher quality images for the high magnification. In Panel C the dots are scattered, on panel D aligned.

Response: We appreciate the reviewer’s comment. We have made some changes, as recommended.

  1. Fig 5: Please replace some of the non-focused or lower quality images with better ones (e.g. CA1 basal, MPTP, day8). Spine density decreased in the MPTP group at day 8 in the CA1 apical, but on day 16 its mean seems to be "restored" to day 1, but the control increased. Were here the mice that were tested in the rotarod experiment used? Same for CA1 basal and DG.

Response: We appreciate the comment. We have replaced the images to show better dendritic spines. As mentioned in the response to the first comment, we set different cohorts for the behavioral test and morphological analysis to restrict any over-reaching effects of the behavioral tests. Following re-analysis using the modified sample size and statistical methods, the reduction effects of MPTP on spine density for each hippocampal region were more clearly observed, with similar levels of controls for each time point.

Discussion:

  1. Dopaminergic innervation of hippocampus is moderate, therefore the rapid changes (within 1 day) is hard to be explained by DA depletion without proper examination - e.g. minimum DA levels by HPLC should be presented and correlated. On the long run, it might be a plausible explanation. Neurons in the HC have most probably very low DAT expression, as this protein is typical for the dopaminergic neurons. Therefore, changes directly due to MPTP is are also not very likely, but they needs further provident. On the other hand, cholinergic innervation of the HC might be decreased after MPTP treatment (Szegö et al., 2013), that can induce long-term changes in the dendritic architecture. Yet, this still does not explain day 1 differences. Changes in the spine density usually occurs earlier than the dendritic complexity, and both have distinct mechanisms. However, the common is that both take longer time.

Response: We appreciate the insightful comment. We have extensively revised the Discussion section.

  1. In addition, although probenecid was found to be neuroprotective in some animal models (e.g. Vamos et al, 2009), probenecid lowers intracellular ATP levels, and can be neurotoxic (Alvarez-Fischer et al., 2013). Therefore, the short-time effect seen in this study might also be attributed to probenecid, and not to DA-dependent effects. Please, discuss this possibility.

Response: Probenecid has been used as an adjuvant with MPTP injection in mice to potentiate neurotoxicity by reducing the clearance of MPTP and its metabolites from the brain and kidneys (Petroske et al., 2001). As commented, Alvarez-Fischer et al. reported the reduced effects of probenecid on intracellular ATP levels; however, they also demonstrated no effect on TH-positive neurons, suggesting that the toxic effects of probenecid affect several neuronal populations apart from dopaminergic neurons. Moreover, other previous studies suggested that probenecid alone has no direct effect on striatal DA content (Lau et al., 1990), indicating that the effects of probenecid itself in our study might be imperceptible. We hope that the reviewer agrees that the focus of our present work is not on this topic.

Reviewer 2 Report

In this manuscript, Weerasinghe-Mudiyanselage et al. have used the acute MPTP PD animal model to study alteration in hippocampal neurons, specifically at CA1 and DG region. Initially, they have semi-quantitatively characterized the model in this study to assess the effect of MPTP injection on motor functioning and TH levels in substantia nigra thereby validating the PD-like phenotype. They further examined the dendritic complexity and spine density of neuronal dendrites in the CA1 and DG region after 1, 8, and 16 days post-MPTP treatment. The manuscript is well-structured, well written, and the claims are well supported in the described results. Even if the mechanistic validation requires more experimentation, the current results are relevant and should be published.

Minor points:

  1. Add statistics to Figure 2B and 4B
  2. It would help orient the readers if authors could add schematic representation of their MPTP model as part of Figure 1.

Author Response

In this manuscript, Weerasinghe-Mudiyanselage et al. have used the acute MPTP PD animal model to study alteration in hippocampal neurons, specifically at CA1 and DG region. Initially, they have semi-quantitatively characterized the model in this study to assess the effect of MPTP injection on motor functioning and TH levels in substantia nigra thereby validating the PD-like phenotype. They further examined the dendritic complexity and spine density of neuronal dendrites in the CA1 and DG region after 1, 8, and 16 days post-MPTP treatment. The manuscript is well-structured, well written, and the claims are well supported in the described results. Even if the mechanistic validation requires more experimentation, the current results are relevant and should be published.

Comment 1. Add statistics to Figure 2B and 4B

Response: We appreciate the reviewer’s comment. As the reviewer 1 commented with regard to the statistical analyses, we have increased the sample size as appropriate to the results from the power analysis computation and re-analyzed the results of all experiments regarding Sholl analysis and dendritic architectures using comparative methods from each animal-based data. Unfortunately, significant differences between groups were not observed at any time point examined. We have revised Figure 2C and 4C for updated results. Additionally, we have added the updated statistical results for the respective Figures in the Supplemental Table 2.

Comment 2. It would help orient the readers if authors could add schematic representation of their MPTP model as part of Figure 1.

Response: As suggested, we have included the schematic diagram for the experimental procedure and cohort in Figure 1A of the revised manuscript to improve clarity.

Reviewer 3 Report

The authors studied the effects of MPTP treatment on hippocampal neural architecture in the Parkinson's Disease model. Overall, the manuscript is very well written. I only have a few minor comments in the discussion:

  1. The authors claim that MPTP treatment inhibits VTA DA neurons which in turn affects hippocampal neurons. However, the authors failed to provide mechanistic explanations, which should be elaborated and further discussed.
  2.  The authors should also discuss potential circuitry mechanisms whereby MPTP might inhibit VTA DA neurons. For example, both lateral habenula and rostromedial tegmental nucleus have been found to inhibit VTA DA neurons (Hikosaka 2007, Jhou 2009, Li 2019a and 2019b), while PPTg and LDT are known to provide excitatory inputs to VTA DA neurons (Lammel 2012, Yau 2016). Potential mechanisms need to be discussed and citations need to be added.

Author Response

The authors studied the effects of MPTP treatment on hippocampal neural architecture in the Parkinson's disease model. Overall, the manuscript is very well written. I only have a few minor comments in the discussion:

 Comment 1. The authors claim that MPTP treatment inhibits VTA DA neurons which in turn affects hippocampal neurons. However, the authors failed to provide mechanistic explanations, which should be elaborated and further discussed.

Response: We appreciate the comment. We have made efforts to explain the possible mechanism related to VTA DA neurons in revised manuscript, as follows:

(Lines 339–372)

“There are several possible mechanisms related to DA underlying this decrease in spine density after acute MPTP treatment in the hippocampus. Firstly, Janakiraman et al. [30] reported reduced dopamine and serotonin levels in the hippocampus after a chronic MPTP regimen. Moreover, dopamine facilitates dendritic proliferation, and DA depletion results in shorter dendrites and reduced spine density in the nucleus accumbens [34], supporting the possibility that prolonged DA depletion in the present study may change the spine density in the hippocampus at the late phase. Moreover, although DA is synthesized mainly by mesencephalic neurons, DA neurons project to the limbic system, including to the hippocampus. Thus, DA receptors are known to control functions related to cognition and synaptic plasticity in the hippocampus [35]. DA innervation in the hippocampus originated in the projections coming from the ventral tegmental area (VTA) [36,37], via this circuitry, DA neurons from the VTA can modulate the plasticity of hippocampal neurons [38]. Thus, acute MPTP may have targeted several possible circuits connecting the hippocampus and VTA, which may include the inhibitory connections of the lateral habenula and rostromedial tegmental nucleus to VTA neurons [39-42], and the excitatory connections of the pedunculopontine tegmental nucleus (PPTg) [43] and laterodorsal tegmentum (LDT) [44] to VTA neurons. This may consequently alter the synaptic plasticity in the hippocampus through changes in the dendritic spine morphology. Further confirmatory studies exploring the roles of these hippocampal-VTA circuitries are needed to uncover the possible underlying mechanisms that lead to the spine density impairment induced by acute MPTP treatment. Moreover, these circuitries may also offer mechanisms by which acute MPTP can elicit non-motor behavioral impairments reminiscent of clinical PD. Furthermore, deficiency in dopamine transporter (DAT) have also resulted to decreased spine densities in the hippocampal CA1 subregion mice [45]. This provides another DA-related mechanism by which acute MPTP altered synaptic densities in the CA1 and DG subregions.

Though present, the DA innervation of the hippocampus is moderate; thus, other non-DA causes of the impairment in synaptic densities must be addressed. For instance, cholinergic signaling may have a role in the synaptic morphological alterations observed, as MPTP treatment have also been reported to induce downregulation of cholinergic markers in the septohippocampal system [46]. These cholinergic deficits, in turn, have been found to reduce dendritic branching and spine density in the neocortical pyramidal neurons of rats [47]. Thus, further studies are needed to verify if cholinergic signaling is involved in the synaptic impairments observed in the current study.”

Comment 2. The authors should also discuss potential circuitry mechanisms whereby MPTP might inhibit VTA DA neurons. For example, both lateral habenula and rostromedial tegmental nucleus have been found to inhibit VTA DA neurons (Hikosaka 2007, Jhou 2009, Li 2019a and 2019b), while PPTg and LDT are known to provide excitatory inputs to VTA DA neurons (Lammel 2012, Yau 2016). Potential mechanisms need to be discussed and citations need to be added.

Response: We appreciate the insightful comment. As mentioned previously, we have extensively modified the Discussion section, including the suggested circuitry mechanisms.

Round 2

Reviewer 1 Report

The authors addressed most of the Questions.

Some concerns remained:

-if the authors used 30 microm section thickness, the distance between sections used for counting should then be 30 or 60 micrometer.

-in the Version what I see, the circles for the Sholl Analysis are very faint, can you replace them?

Author Response

Responses to the Reviewers’ Comments (Round 2)

 The authors addressed most of the Questions.

Some concerns remained:

Comment 1. If the authors used 30 microm section thickness, the distance between sections used for counting should then be 30 or 60 micrometer.

Response: We sincerely appreciate the reviewer’s comment. We apologize for the inappropriate detail in the methodology. We have corrected the distance by which the non-overlapping sections for TH-immunohistochemistry was chosen as 60 µm in the revised manuscript.

We have revised the text regarding TH staining to offer more correct information as follows:

(Line 142)

“The numbers of TH-immunopositive cells from the three non-overlapping sections (approximately 60 µm apart) were averaged per animal.”

Comment 2. In the Version what I see, the circles for the Sholl Analysis are very faint, can you replace them?

Response: We sincerely appreciate the reviewer’s comment. Authors agree with the reviewer’s comment that the concentric circles in the figures are difficult to discern. Therefore, in the revised version we tried our best to increase the thickness of the circles for the Sholl.
